# Revisiting GNNs for Boolean Satisfiability

## Abstract

We introduce a number of enhancements for the training and inference procedure of Graph Neural Networks that are trained to predict solutions of combinatorial problems. We motivate these enhancements by pointing to possible connections to two approximation algorithms studied in the domain of Boolean Satisfiability: Belief Propagation and Semidefinite Programming Relaxations. The first significant enhancement is a curriculum training procedure, which incrementally increases the problem complexity in the training set together with increasing the number of message passing iterations of the Graph Neural Network. We show that the curriculum, together with several other optimizations, reduces training time by more than an order of magnitude compared to the baseline without the curriculum. Furthermore, we apply decimation and initial embedding sampling, which significantly increase the percentage of solved problems.

## 1 Introduction

This work advances the training and inference procedure of Graph Neural Networks (GNNs) used in the context of combinatorial problems (Cappart et al., 2023; Gasse et al., 2019; Lamb et al., 2020). We target Boolean Satisfiability (SAT), which can be seen as a prototypical combinatorial problem, with many practical applications. There we demonstrate a substantial increase in the training efficiency and the number of solved problems. The presented improvements are motivated by two well-studied approximation algorithms, which could potentially shed light on the inner workings of the trained model.

Previously, GNNs have already been shown to learn to predict the satisfiability status of CNF formulas (Selsam et al., 2019). Although these learned solutions lag far behind the solvers used in practice in terms of the size of problems they are able to solve, it is desirable to understand the process by which they arrive at their outputs. Moreover, neural networks can potentially find solutions (in our case, approximation algorithms), which could lead to unexpected insight (Pickering et al., 2023; Davies et al., 2021), such as discovering new heuristic algorithms by interpreting the learned weights in the neural network.

It is natural to try to interpret the message passing (MP) process of a GNN as an optimization of a continuous relaxation of the original problem. Recently, Kyrillidis et al. (2020) demonstrates scenarios where solving a continuous relaxation formulation may provide benefits over solving the formula using standard solvers (i.e. having a better performance for non-CNF formulas). This suggests promising future applications of GNN-based solutions. GNN-based solutions could potentially bring improvements over manually designed continuous solvers because they can adapt to the specifics of a given distribution of problems. Also, having a better understanding of learned solutions can eventually lead to a confluence of continuous solvers and graph neural networks, which would be an analog of Physics-Informed Neural Networks (Raissi et al., 2019) in the context of combinatorial problems.

When treating the message-passing process of a GNN as an optimization of a continuous relaxation of the original problem, the relaxed variables are in this context represented by high-dimensional vectors. This hints at a possible connection to *Semidefinite Programming (SDP)* based approximation algorithms. The solving process of an SDP solver can be understood as an incremental optimization of a set of vectors representing the variables being optimized. After convergence, these vectors are rounded to give a solution to the original discrete problem.

Another way of interpreting the MP process of a GNN is through the lens of *Belief Propagation* algorithms (Park & Shin, 2014). In the context of Boolean Satisfiability, Belief Propagation algorithms operate on similar literal-clause factor graphs as a GNN. A particular version called *Survey Propagation* also sends messages from literals to clauses in the form of 3-dimensional vectors. In the domain of *random satisfiability*, these algorithms have been proven very effective (Braunstein et al., 2005) and their theoretical properties are well-studied.

In this work, we empirically test whether these connections could give us novel insights into how trained GNNs operate and bring about improvements in terms of their training speed and accuracy. To our knowledge, these connections have not been explored before. We also emphasize that we do not derive any formal statements regarding these connections. We demonstrate that the behavior of the trained GNN is qualitatively similar to these approximation algorithms and that using insights from these approximation algorithms can lead to significant improvements.

We present the following novel contributions:

- We demonstrate several similarities between the empirical behavior of trained GNN and two well-studied approximation algorithms for Boolean Satisfiability.
- Motivated by these connections, we design a training curriculum that speeds up the training process by an order of magnitude.
- For a trained model with fixed weights, we propose to sample different initializations of literals and to apply a decimation procedure (inspired by Belief Propagation). This substantially increases the number of solved problems (problem is considered solved if the network correctly predicts satisfiablity *and* produces a satisfying assignment for satisfiable instances).

In Section 2, we provide relevant background information; Sections 3 4 describe our contribution; Section 5 contains experimental results and is followed by a conclusion.

## 2 BACKGROUND

### 2.1 BOOLEAN SATISFIABILITY

Basic background knowledge of propositional logic is assumed, cf. Biere et al. (2021). Boolean variables are denoted by $x_1, x_2, \ldots$; disjunction by $\vee$, conjunction by $\wedge$, and negation by $\neg$. A *literal* is a variable or its negation; a clause is a disjunction of literals. For a literal $l$ we write $\bar{l}$ for the *complementary literal* of $l$, i.e. $\bar{x}$ is $\neg x$ and $\overline{\neg x}$ is $x$. A formula in *conjunctive normal form (CNF)* is a conjunction of clauses. Whenever convenient, a clause is treated as a set of literals and a CNF formula as a set of sets of literals. A clause is called *unit* iff it consists of a single literal.

An *assignment* is a total mapping from variables to $\{0, 1\}$, representing true/false. An assignment $\sigma$ is *satisfying* a formula $\phi$ iff $\phi$ evaluates to true under the standard semantics of Boolean connectives. In particular, an assignment satisfies a CNF $\phi$ iff it satisfies at least one literal in each clause.

There exist multiple representations of CNF formulas in the form of graph. In this work, we use the *literal-clause factor graph*, which is an undirected bipartite graph of clauses and literals. Each node of a literal in this graph is connected to nodes of clauses that contain this literal.

*MaxSAT* is an optimization version of SAT where one is given a CNF $\phi$ and the objective is to find a variable assignment that maximizes the number of satisfied clauses. So, for instance, in $\{x_1 \vee x_2, \neg x_1, \neg x_2\}$ the assignment $x_1 = 1, x_2 = 0$ satisfies the first 2 clauses but not the last one, and it is the optimum because the 3 clauses cannot be satisfied simultaneously. The problem is NP-hard even for formulas that have only 2 literals in each clause (Papadimitriou, 1994).

### 2.2 RANDOM SATISFIABILITY AND MESSAGE-PASSING ALGORITHMS

Random satisfiability provides a natural and simplified setting to study the computational hardness of finding a satisfying assignment and the structure of the space of satisfying assignments. Typically, it is assumed that each clause in the formula is sampled randomly and has the same number of variables ($k$). A random formula is parametrized by a parameter $\alpha$ denoting the clause-to-variable

ratio. As the problem size increases asymptotically, the solution space undergoes several phase transitions as the parameter $\alpha$ changes. When random clauses are added to the formula, it becomes increasingly challenging to find a satisfying assignments until it reaches a point where the formula becomes unsatisfiable. This occurs at the *satisfiability threshold* whose value for $k > 2$ is known only through upper and lower bounds and numerical estimates (Zdeborová & Krzakala, 2016).

Before reaching the phase transition to unsatisfiability, the geometry of the solution space undergoes several other phase transitions, during which the set of solutions breaks into well-separated clusters (in terms of the Hamming distance) (Kroc et al., 2012). Each cluster corresponds to a set of solutions in which specific variables are fixed (to a value 0 or 1) and the values of the remaining variables could vary (this is denoted by the value $*$).

Unlike real-world formulas, random formulas could be efficiently solved by message-passing algorithms (Braunstein et al., 2005). These algorithms could be viewed as algorithms that compute the marginal probability of individual variables using a belief propagation algorithm (BP) (Kroc et al., 2012). The marginal probability that a variable $x_i$ will have a value 1 in a satisfying assignment is given by a proportion of assignments where $x_i = 1$ among all possible satisfiable assignments. Belief propagation can compute these marginal probabilities exactly for formulas whose factor graphs are devoid of loops. Clearly, computing these marginal probabilities exactly for a generic formula is much harder than finding a single satisfying assignment. For factor graphs with loops, BP can be regarded as an algorithm that tries to approximate these marginal probabilities (Maneva et al., 2007).

To obtain an algorithm for finding a satisfying assignment with BP, one can employ a *decimation* procedure in which variables with the most extreme estimated marginal values are fixed[1] and the whole process is repeated by running the BP process again on the reduced formula until no variable has a sufficiently high estimated marginal. At this point, the resulting formula would be solved by a local search. Empirically, it has been observed that BP starts to fail as the parameter $\alpha$ approaches the satisfiability threshold. This phenomenon is frequently attributed to the evolving structure of the solution space (Maneva et al., 2007). As the parameter $\alpha$ increases, long-range correlations between variables start to appear[2] and this breaks the assumption needed for BP to work properly.

Braunstein et al. (2005) overcome this problem by drawing on concepts from statistical physics to design an algorithm called *Survey Propagation* (SP). It can be viewed as a BP running on an augmented factor graph of the original formula in which each variable can take one of three values: $0, 1, *$. SP was empirically shown to find satisfying assignments of problems with parameter $\alpha$ very close to the satisfiability threshold. Similarly to BP, SP works by iteratively sending messages on a literal-clause factor graph until the convergence threshold is reached. Unlike BP, the messages from literals to clauses have the form of 3-dimensional vectors expressing the marginals for the three possible values.

## 2.3 SEMIDEFINITE PROGRAMMING FOR BOOLEAN SATISFIABILITY

Semidefinite programming (SDP) is a mathematical optimization technique that is primarily used for problems involving positive semidefinite matrices. In SDP, a linear objective function is optimized over a feasible region given by a *spectrahedron* (an intersection of a convex cone formed by positive semidefinite matrices and an affine subspace) (Ramana & Goldman, 1995). Along with the broad scope of applications, SDP has also been used to design approximation algorithms for discrete NP-hard problems (Gärtner & Matousek, 2012). This is achieved by lifting variables of a problem to a vector space and optimizing a loss function expressed in terms of these vectors. Here we illustrate this process on a Semidefinite Relaxation of a MAX-2-SAT problem.

MAX-2-SAT is a version of MAX-SAT in which each clause contains at most two literals. The semidefinite relaxation of a MAX-2-SAT problem can be formulated as follows (Gomes et al., 2006): To each Boolean variable $x_i$ (where $i \in \{1, 2, \ldots, n\}$), a new variable $y_i \in \{-1, 1\}$ is associated, and an additional variable $y_0$ is introduced (this variable can be understood as representing the value *true*). By definition, $x_i$ is true if and only if $y_i = y_0$, otherwise, it is false. Using these new variables, we can represent each clause by an expression that is maximized when the clause is

---

[1] A variable is fixed if its estimated marginal reaches a given threshold.

[2] Variables that are far apart in the factor graph start to be correlated, i.e. knowing the value of variable $x_i$ in a satisfying assignment provides information about the value of the variable $x_j$

satisfied (considering only values in $\{-1, 1\}$). Each expression contains binary products between variables used in the given clause (see Appendix A.2). By summing the expressions of all the clauses in the formula, we obtain a quadratic objective function that gives a maximal value when the maximum number of clauses is satisfied. Therefore, the whole problem may be stated as an integer quadratic program where the constraints restrict the values of the variables to $\{-1, 1\}$.

The SDP relaxation is obtained by lifting each variable $y_i$ to a $(n+1)$-dimensional unit vector $\mathbf{y_i}$. Therefore, the binary products $y_i \cdot y_j$ in the objective function are replaced by inner products $\langle \mathbf{y_i}, \mathbf{y_j} \rangle$. This can be compactly represented in matrix form if we substitute each inner product $\langle \mathbf{y_i}, \mathbf{y_j} \rangle$ by a scalar $Y_{ij}$ of a matrix Y. The fact that these scalars correspond to inner products could be encoded by the restriction to *positive-semidefinite* matrices Y. We can thus represent the original MAX-2-SAT problem as the following SDP:

$$\textbf{Maximize: } \text{Tr}(WY) \quad \textbf{Subject to: } Y_{ii} = 1 \text{ for all } i \in \{0, 1, \dots, n\} \quad Y \succeq 0,$$

where Tr denotes the trace of a matrix. Both $Y$ and $W$ are $(n+1) \times (n+1)$ matrices. Matrix $W$ is a coefficient matrix of the objective function, derived from the clauses. More detailed derivation is available in appendix A.1.

Positive semidefinitness of matrix $Y$ assures that the matrix can be uniquely factorized as $Y = Y^{\frac{1}{2}} Y^{\frac{1}{2}}$. Rows of the matrix $Y^{\frac{1}{2}}$ are real vectors $\mathbf{y}_i$ for all $i \in \{0, \dots, n\}$ and values in the original matrix $Y_{ij}$ are their inner products $\langle \mathbf{y_i}, \mathbf{y_j} \rangle$ for all $i, j \in \{0, \dots, n\}$. The constraints $Y_{ii} = 1$ assures that all vectors $\mathbf{y}_i$ lie on $(n+1)$ dimensional unit sphere.

The solver for this SDP optimizes the numbers in the matrix $Y$ but using the factorization, we can possibly visualize what happens with the vectors $\mathbf{y_i}$. The process starts with random unit vectors which are continuously updated in order to maximize the objective function. If we would further fix the position of the vector $\mathbf{y_0}$ (corresponding to the value *true*) we would see that the vectors of variables that will be set to true in the final assignment are getting closer to the vector $\mathbf{y_0}$ and the vectors $\mathbf{y_j}$ of variables that will be set to false will be moving away from it so that the inner product $\langle \mathbf{y_0}, \mathbf{y_j} \rangle$ is close to $-1$. If the formula is satisfiable, the objective function drives the vectors to form two well-separated clusters. However, if only a few clauses could be satisfied at the same time, the vectors would end up being scattered.

A simple way to round the resulting vectors $(\mathbf{y}_1, \dots \mathbf{y}_n)$ and get the assignment for the original Boolean variables is to compute an inner product $\langle \mathbf{y_0}, \mathbf{y_i} \rangle$ and assign the value according to its sign. It is also possible to assign the values by picking a random separating hyperplane and it can be shown that this rounding gives $0.8785$-approximation of the integer program optimum (Goemans & Williamson, 1995). Similar SDPs can be obtained for different versions of MAX-SAT (with larger clauses). From an empirical observation, the convergence threshold of the SDP solver needs to be decreased significantly compared to MAX-2-SAT in order to obtain a good approximation for these more complicated versions, which is related to our curriculum training procedure.

We mention that the expressions of the clauses reach their maximum at 1 (when a clause is satisfied by the assignment). This means that the whole formula is satisfiable if the objective function achieves a value that is equal to the number of clauses in the formula. Therefore, we can theoretically obtain an incomplete SAT solver from this SDP.

In Section 4, we empirically demonstrate that the behavior of a trained GNN resembles the optimization process described above. With this intuition, we propose several improvements that lead to faster training time and higher accuracy.

## 2.4 Graph Neural Networks for Boolean Satisfiability

GNNs constitute a flexible tool for learning representations of graph-structured data. Representing the input data in the form of a graph allows one to encode complex relations and sparsity structures. GNNs then allow to encode inductive biases such as invariance to various transformations (Bronstein et al., 2021). For these reasons, GNNs are frequently used in applications of machine learning to combinatorial optimization (Gasse et al., 2019; Lamb et al., 2020; Cappart et al., 2023) where optimization problems are often amenable to graph-based representations.

Typically, a GNN would enhance a manually designed solver by replacing various decision heuristics with their predictions after being trained either in a supervised or reinforcement learning mode (Bengio et al., 2021; Gasse et al., 2019). Another area of research focuses on end-to-end approaches where the GNN is trained to produce the final answer (Selsam et al., 2019). From a practical point of view, these end-to-end approaches are interesting because they can potentially find more efficient solutions than those proposed by algorithm designers (Veličković & Blundell, 2021).

As other data-driven algorithms, GNNs used for combinatorial optimization make a trade-off between performance on some restricted subset of inputs and generalization to the whole domain of possible inputs. In the extreme case, the input distribution may be skewed to the extent that the GNN only needs to recognize superficial features of the input graph.

In this work, we focus on the end-to-end approaches. We demonstrate these improvements with the popular NeuroSAT architecture (Selsam et al., 2019) which has demonstrated the ability to exhibit nontrivial behavior resembling a search in a continuous space, rather than mere classification based on superficial statistics.

**The NeuroSAT architecture** We demonstrate our enhancement using the NeuroSAT architecture with several simplifications. NeuroSAT is a GNN that operates on an undirected bipartite graph of literals and clauses. In this graph, each literal is connected to clauses that contain this literal. The MP process alternates between two steps that update the representations of clauses and literals, respectively. The embeddings of literals and clauses are updated by two independent LSTMs. The messages from clause to literals are produced with a 3-layer MLP that takes the embeddings of a clause as an input, and similarly in the opposite direction. After several rounds of MP iterations, the vector representation of each literal is passed into another MLP used to produce a vote for the satisfiability of the formula. These votes are averaged across all literals to produce the final prediction. A more detailed description is provided in Apendix A.2.

## 3 CURRICULUM FOR TRAINING GNNS

An important feature of the NeuroSAT architecture is that the number of MP iterations does not need to be fixed because each round of MP is realized with the same function. In the original paper, the authors demonstrated that the model trained with 26 MP iterations on problems with up to 40 variables was able to generalize to much larger problems with up to 200 variables. This is achieved just by iterating MP for more steps (hundreds or even thousands). Therefore, we can view the architecture as an iterative algorithm with an adaptive number of steps that could depend on the difficulty of the problem. During training, the number of iterations needs to be fixed so that the problems can be solved in batches, but during inference, each problem can run for a different number of steps.

As was already shown in the original paper, when the problem is satisfiable and the model correctly predicts it, the vectors of literals form two well-separated clusters. Empirically, once the vectors form two well-separate clusters, subsequent updates do not change the vectors significantly. Informally speaking, MP iterations can be viewed as optimization steps of an implicit energy function of the trained model (Gould et al., 2021). Unsatisfied clauses should increase the energy and the minimum energy should be achieved when the maximum number of clauses are satisfied. For satisfiable formulas, this happens when the vectors form two well-separated clusters, which makes the whole process qualitatively similar to the optimization of the SDP relaxation described in Section 2.3 (for supportive arguments of this claim, see Appendix A.5). Therefore, we can set up a stopping criterion that stops the MP process once the vectors stop to change significantly. This could be viewed as an analog of a convergence threshold of iterative solvers for continuous problems or BP.

As mentioned in Section 2.3, the number of iterations required is well correlated with the difficulty of the problem. This motivates our curriculum training procedure, which trains the model by incrementally enlarging the training set with bigger problems and increasing the number of MP operations. For each new problem size, the model is trained until a certain accuracy is reached, and after that, larger problems are added to the training set and the number of MP rounds is incremented accordingly. With this procedure and several simplifications of the original model, we achieve almost an order of magnitude faster convergence to the same accuracy as reported in the original paper (85%).

A similar observation was recently made by Garg et al. (2022) in a study of the in-context learning capabilities of a trained transformer. The authors observe that the trained model is able to perform optimization of a loss function (a high-dimensional regression) as the input passes through individual layers. They also experimentally demonstrated that it is possible to significantly accelerate the emergence of this capability if the model is trained incrementally by increasing the dimensionality of the regression. In our case, we incrementally increase also the number of MP iterations together with the number of variables within the formula, which speeds up the training even further.

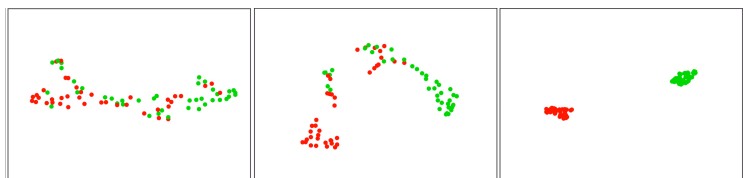

Figure 1: This figure shows how the embeddigs of literals change during the MP process. We selected 3 different time steps from the 30 time steps used for this example. The 16-dimensional vectors are projected to 2D by UMAP algorithm. The colors correspond to the truth values of the final solution recovered for this formula. As can be seen, the literals progressively form two well-separated clusters of literals with the same truth value.

## 4 SAMPLING AND DECIMATION

Selsam et al. (2019) observes that for formulas that the model correctly classified as satisfiable, the embeddings of literals form two well-separated clusters. In Figures 1 and 5, we recapitulate their visualization of embeddings with UMAP instead of PCA (the final clusters are more distinct when visualized with UMAP). The authors showed that for a large portion of correctly classified satisfiable formulas, they were able to recover a satisfying assignment by clustering the embeddings and by assigning the same Boolean value to all literals within one cluster. They needed to test both possible ways of assigning Boolean values because they did not know in advance which cluster corresponds to the value *true* and which to the value *false*.

We first confirmed their finding by running an experiment in which we removed the final voting layer and classified the formula using a *Silhouette score* (Rousseeuw, 1987) of the embeddings. Concretely, we first run K-means on the embeddings of literals to assign them to two clusters and then we compute the Silhouette score with the assigned labels. On the training set, we estimate a threshold for this score and classify the test set according to this threshold (i.e., we classify a formula as satisfiable if its score is above this threshold). With this procedure, we achieve the same accuracy as the original model with the voting layer (85%), which means that the observation of cluster formation is robust.

Next, we tested whether the model can produce different clusterings if we sample initial embeddings of literals multiple times (this corresponds to different initializations of the SDP solver). This turned out to be the case; for a large portion of satisfiable problems, using multiple random initializations of the embeddings would produce a diverse set of solutions. *This enables us to substantially improve the classification accuracy by taking a majority vote over multiple initializations.* Sampling different solutions is also trivially parallelizable.

Motivated by similarities with the SDP relaxation, we tested whether it is possible to recover the vector representing the value *true* and use this vector to assign value to each literal. We selected all the formulas that the model correctly classified as satisfiable and checked whether the formula could be satisfied by one of the two possible assignments of Boolean values to the resulting clusters. Thus, we obtain a set of literal embeddings that correspond to the value *true* and another set corresponding to the value *false*, aggregated over all problems. Finally, we computed an average vector for both sets and also a distribution of Euclidean distances to these two vectors. Concretely, for each literal that was assigned the value *true*, we compute the distance of its embedding to the average *true* vector and also to the average *false* vector. Similarly, for each literal that was assigned the value *false*.

Figure 2 shows that all vectors assigned to the value *true* are close to the average *true* vector and far from the average *false* vector, and vice versa. Therefore, we can assign each literal of a formula

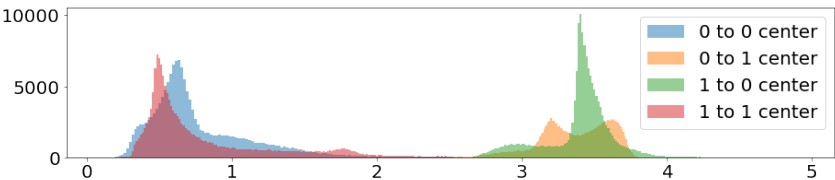

Figure 2: A histogram of Euclidean distances to the average *true* vector and average *false* vector. **0 to 0 center** are distances between embeddings of literals assigned to *false* to the average *false* vector, etc. The figure clearly demonstrates that literals that take the value *false* in the final assignment move to the same area of the vector space and the same is true for the opposite polarity.

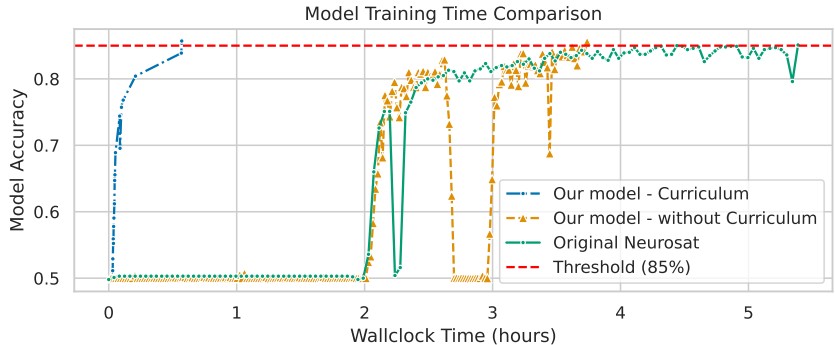

Figure 3: Validation accuracy during training. Our model with a curriculum achieves reaches 85% in approximately 30 minutes, whereas the original NeuroSAT implementation needs over 5 hours. For comparison, we also add our implementation trained on the same data but without a curriculum. The training of each model stops once it achieves an accuracy of 85% on a validation set.

according to its distance to these two average vectors. Relying on the intuition from BP, we may try to treat these two distances as marginal probabilities over the two possible truth values, and therefore obtain a decimation algorithm. This algorithm fixes variables whose embeddings are close to one of these average vectors, simplifies the formula, and runs the MP of the GNN again. The results with these improvements are presented in the following section.

## 5 EXPERIMENTS

**Data Generation** Selsam et al. (2019) demonstrates that a well-structured distribution of training data is essential to prevent the model from overfitting to superficial features. A recent theoretical study (Raventós et al., 2023) also explains why input diversity is important in order for the model to transition to a regime where it is performing optimization over inputs. We therefore reuse Selsam's generative model of random SAT formulas which makes sure that no superficial features exist. However, Selsam's generation procedure is largely random and therefore does not capture any human-like reasoning skills. Therefore we also generate structured problems (Latin squares, Sudoku, and logical circuits). The details of data generation are described in Appendix A.3.1 and A.3.2

In the reference implementation of NeuroSAT, the model is trained on 100,000 formulas where for each formula the number of variables is sampled uniformly from the interval $[10, 40]$. The model is then evaluated on problems with 40 variables (similarly as in the orinal paper, this test set is here referred to as $SR(40)$). In our case, the size of the test set is the same, but we train on 10,000 formulas in total and sample the number of variables in the formula from the interval $[5, 40]$.

**Model architecture** When experimenting with the original NeuroSAT architecture, we found that it is unnecessarily complex. We managed to significantly simplify the model without sacrificing the final accuracy. Here is the list of simplifications in our model:

- We completely removed the two 3-layer MLPs that produce the messages from the hidden states of the two LSTMs. The messages sent are, therefore, the hidden states themselves.
- We replace the final voting MLP with a linear layer.
- We do not use LayerNorm within LSTMs.
- We reduce the dimension of the hidden state of the LSTMs from 128 to 16.

**Training loop** We train the model using the curriculum described in Section 3. In the following text, we consider the size of the formula to be given by the number of variables it contains. The training starts with formulas of size 5 and this size is incremented by 2 every time the validation accuracy (for a given size) reaches a given threshold or the maximum number of 200 epochs is reached. For each increment, we add the problems from the four previous increments [3] which makes the training more stable. The thresholds used to increment the size are obtained by interpolating the values between $0.65$ (for the first size) and $0.85$ (for the last size). We note that the values could be set to a fixed number but this may waste time during learning on the intermediate sizes. Empirically, the model spends most of the time on the first 3 and 5 last sizes. For the other training hyperparameters, we follow the original implementation except that we change the learning rate to $2 \cdot 10^{-3}$.

## 5.1 RESULTS

**Training Convergence** To demonstrate the effectiveness of the proposed curriculum, we compare the training process with two baselines. The first is the publicly available implementation of NeuroSAT [4] and the second is our model without the curriculum. We stop training each model once it reaches the validation accuracy reported in the original paper (85%). As visible in Figure 3, our model with the curriculum reaches this accuracy in approximately 30 minutes, while the other two baselines need to be trained for several hours[5]. All models were trained on 1 GPU (NVIDIA A100).

**Sampling and Decimation** In Table 1, we show the increase in accuracy due to the enhancements described in Section 4. Together with the results on randomly generated problems, we also show results on three different structured problems whose details are described in the Appendix A.3.2. The results show a noticeable increase in the number of solved problems for both enhancements (sampling and decimation). For decimation, we use only two passes, which means that if the first application of the GNN did not solve the formula, we fix the variables whose distances to the average vectors were below a threshold[6], simplify the formula, and then process it with the GNN again. For each subsequent pass, we used only one initialization sample for each decimated formula. Therefore, if the first pass uses 16 samples, the second pass can also produce a maximum of 16 samples. To see the effects of decimation, we show results of runs with the same number of samples in total (32) but without decimation.

For results with 3 passes, see Appendix A.6. We note, as should be obvious, that our method cannot certify unsatisfiability.

## 6 RELATED WORK

Most of the related work was already mentioned in Section 2. In this section, we describe related work in the context of GNNs and Boolean Satisfiability.

In the domain of Boolean Satisfiability, applications of GNNs can be divided into hybrid or end-to-end approaches. In the hybrid approaches, the GNN is used to guide a discrete search. We can further distinguish between applications where the GNN guides a simple heuristic and applications where the predictions of the GNN are used inside an industrial SAT solver. For the case heuristics, Yolcu & Póczos (2019) use GNNs trained by Reinforcement Learning to select variables and values in a local search. Zhang et al. (2020) also use GNN for local search, but train it with supervised learning.

---

[3] That is, for the formulas of size 21, we add formulas of size 19, 17, 15, 13.

[4] Available at this url: `https://github.com/ryanzhangfan/NeuroSAT/tree/master`.

[5] The precise numbers are: 34 min for our model with a curriculum, 5h 23min for the original NeuroSAT implementation, and 3h 44min for our model without a curriculum.

[6] We set this threshold to 1.9 after a visual inspection of Figure 2

| Problem type | #SAT problems | Avg. #var | First pass | Second pass | #MP iterations | #samples | Decimation | Solved |
|---|---|---|---|---|---|---|---|---|
| **SR(40)** | 5000 | 40 | 4442 | 274 | 100 | 16 | Yes | 94 % |
| | | | 3990 | - | | 1 | No | 80 % |
| | | | 4457 | - | | 32 | No | 89.1 % |
| **Latin Squares 9x9** | 200 | 196.9 | 186 | 14 | 1000 | 16 | Yes | 100 % |
| | | | 95 | - | | 1 | No | 47.5 % |
| | | | 192 | - | | 32 | No | 96 % |
| **Latin Squares 8x8** | 200 | 133.5 | 196 | 1 | 1000 | 16 | Yes | 98.5 % |
| | | | 113 | - | | 1 | No | 56.5 % |
| | | | 197 | - | | 32 | No | 98.5 % |
| **Logical Circuits** | 344 | 131.1 | 319 | 0 | 1000 | 16 | Yes | 92.7 % |
| | | | 293 | - | | 1 | No | 85.2 % |
| | | | 319 | - | | 32 | No | 92.73 % |
| **Sudoku 9x9** | 200 | 245.6 | 92 | 11 | 1000 | 16 | Yes | 51.5 % |
| | | | 35 | - | | 1 | No | 17.5 % |
| | | | 94 | - | | 32 | No | 47 % |

Table 1: Improvements obtained due to the sampling and decimation procedure. We test how many satisfiable problems could be solved with the model. **SR(40)** is the test set with random problems; the other prolems are describe in Appendix A.3.2. For each problem type, we include the **average number of variables**. For larger problems, we run the model for more MP iterations. When decimation is used, we count the number of problems solved during the **first and the second pass** separately. **#samples** refers to the number of different initializations of literal embeddings.

For the case of SAT solvers, Kurin et al. (2020) introduce a branching heuristic for SAT solvers trained using value-based reinforcement learning (RL) with GNNs for function approximation. They incorporate the heuristic with the MiniSat solver and manage to reduce the number of iterations required to solve SAT problems by 2-3X. Similarly, Wang et al. (2021) use GNN as a variable selection heuristic and manage to improve MiniSat in terms of the number of solved problems on the SATCOMP-2021 competition problem set.

On the end-to-end front, the most relevant work is the one by Selsam et al. (2019) who introduced the NeuroSAT architecture, which was our starting point. Similar to NeuroSAT were the models introduced by Cameron et al. (2020) who used different GNN architecture and Shi et al. (2022) who used a *Transformer*. Freivalds & Kozlovics (2022) use a *Denoising Diffusion* model to learn to sample multiple solutions and Ozolins et al. (2022) propose an approach in which the GNN can take feedback from solution trials.

Apart from work focused on Boolean Satisfiability, we also mention the work by Kuck et al. (2020) who use GNN to improve Belief Propagation.

## 7    CONCLUSION

We revisit and enhance the training and inference procedure of GNNs used in the context of combinatorial problems. In particular, we focus on the well-known NP-complete problem of Boolean Satisfiability (SAT). We introduce a curriculum training procedure, which enables a significantly faster iteration over experiments. Further, we apply a decimation procedure and initial-value sampling, which significantly increase the number of solved problems. For a problem to be considered solved, we not only require the correct prediction whether it is satisfible or not, but we also require the GNN to produce a satisfying assignment for satisfiable problem instances.

The introduced enhancements are motivated by noticing similarities between the behavior of a trained GNN and two well-studied approximation algorithms. Even though the enhancements were presented in the domain of Boolean Satisfiability, we believe that they can easily be generalized to other domains where these approximation algorithms are used. In future work, we plan to explore these similarities more closely and reverse engineer the algorithm learned by the GNN. We also hope that our work could eventually shed light on the connection between SDP and BP.

## 8 REPRODUCIBILITY

In the supplementary material, we provide a source code with a readme that allows to reproduce the results of our model presented in Figure 3 and Table 1. Results for NeuroSAT presented in Figure3 were obtained by running training scripts in the publically available implementaion of NeuroSAT available at this url:`https://github.com/ryanzhangfan/NeuroSAT/tree/master`.

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

# A    APPENDIX

## A.1    DERIVATION OF THE SDP RELAXATION FOR MAX-2-SAT

Here we provide further details about the definition of SDP relaxation for MAX-2-SAT. The goal is to write an objective function for 2-CNF formulae, which consist of clauses $c_1, \ldots, c_k$ over variables $x_1, \ldots, x_n$ with at most two literals per clause.

For each Boolean variable $x_i$ (where $i \in \{1, 2, \ldots, n\}$) a new variable $y_i \in \{-1, 1\}$ is first instantiated and one additional variable $y_0 \in \{-1, 1\}$ is introduced. The additional variable is introduced to unambiguously assign the truth value in the original problem from values of relaxed problem. It is not possible to just assign *True* (*False*) to $x_i$ if $y_i = 1(0)$ because quadratic terms cannot distinguish between $y_i \cdot y_j$ and $(-y_i) \cdot (-y_j)$. Instead, the truth value of $x_i$ is assigned by comparing $y_i$ with $y_0$: $x_i$ is *True* if and only if $y_i = y_0$ otherwise it is *False*. The assignment is therefore invariant to negating all variables.

To determine the value of a formula, we sum the value of its clauses $c$ which are given by the value function $v(c)$. Here are examples of the value function for 3 different clauses:

$$v(x_i) = \frac{1 + y_0 \cdot y_i}{2}$$
$$v(\neg x_i) = 1 - v(x_i) = \frac{1 - y_0 \cdot y_i}{2}$$
$$v(x_i \vee \neg x_j) = 1 - v(\neg x_i \wedge x_j)$$
$$= 1 - \frac{1 - y_0 \cdot y_i}{2} \frac{1 + y_0 \cdot y_j}{2}$$
$$= \frac{1}{4}(1 + y_0 \cdot y_i) + \frac{1}{4}(1 - y_0 \cdot y_j) + \frac{1}{4}(1 + y_i \cdot y_j)$$

By summing over all clauses $c$ in in the Boolean formula, the following integer quadratic program for MAX-2-SAT is obtained:

$$\text{Maximize:} \quad \sum_{c \in C} v(c)$$
$$\text{Subject to:} \quad y_i \in \{-1, 1\} \text{ for all } i \in \{0, 1, \ldots, n\},$$

this can be rewritten by collecting coefficients of $y_i \cdot y_j$ for $i, j \in \{0, 1, \ldots, n\}$ and putting them symmetrically into a $(n+1) \times (n+1)$ coefficient matrix $W$. The terms $y_i \cdot y_j$ can be collected in a matrix $Y$ with same dimensions as $W$. The elements $Y_{ij}$ correspond to $y_i \cdot y_j$ for $i, j \in \{0, 1, \ldots, n\}$. Both matrices are symmetric, hence the sum of all elements in their element-wise product (which is the objective function) can be compactly expressed by using trace operation. This leads to the following version of the same integer program:

$$\text{Maximize:} \quad \text{Tr}(WY)$$
$$\text{Subject to:} \quad Y_{ij} \in \{-1, 1\} \text{ for all } i, j \in \{0, 1, \ldots, n\}, i \neq j.$$

So far no relaxation has been made. To make the discrete program continuous, the value of the variables $y_i$ is allowed to be any real number between $-1$ and $1$. After solving a quadratic program with this relaxation, rounding can be used to obtain a value from $\{-1, 1\}$.

Semi-definite programming goes further and allows variables to be $(n+1)$-dimensional unit vectors $(y_0, \dots y_n) \longrightarrow (\mathbf{y}_0, \dots \mathbf{y}_n)$. schematicaly depicted in figure 4. This directly leads to the relaxation used in the main part of this study.

Our aim was to show that solving SDP relaxation by optimization and rounding by separating the high-dimensional vectors closely resembles the behavior of GNN.

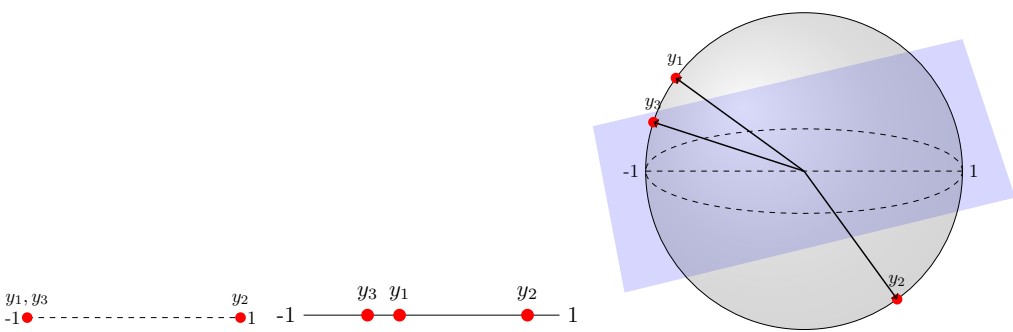

Figure 4: Lifting the variables to a higher dimension, demonstrated on variables $y_1, y_2, y_3$. Initially, only integer values of $-1$ and $1$ could be assigned to them (integer program). Next, constraints are relaxed, allowing variables to take any real value between $-1$ and $1$. Finally, it is permitted for them to be unit vectors in a high-dimensional space (here, 3 dimensions). The hyperplane in the last picture would be used for rounding the variables at the end. This hyperplane can be randomly selected, and truth values for variables $y_1, y_2, y_3$ are determined based on which side of the hyperplane they land after continuous optimization.

## A.2 THE NEUROSAT ARCHITECTURE

For completeness, we provide the update rules and voting rule from the original paper Selsam et al. (2019):

$$(C^{(t+1)}, C_h^{(t+1)}) \leftarrow \mathbf{C_u}([C_h^{(t)}, M^\top \mathbf{L_{msg}}(L^{(t)})]) \tag{1}$$

$$(L^{(t+1)}, L_h^{(t+1)}) \leftarrow \mathbf{L_u}([L_h^{(t)}, \text{Flip}(L^{(t)}), M\mathbf{C_{msg}}(C^{(t+1)})]) \tag{2}$$

$$L_*^T \leftarrow \mathbf{L_{vote}}(L^{(T)}) \in \mathbb{R}^{2n}. \tag{3}$$

The first rule is used to update the clause embedding matrix $C^{(t)} \in \mathbb{R}^{m \times d}$ where $d$ is the size of the hidden feature vector and $m$ is the number of clauses, $t$ is a discrete time step. The second rule is used to update literals whose embedding are stored in matrix $L^{(t)} \in \mathbb{R}^{2n \times d}$, where $n$ is number of variables (there are $2n$ rows to cover both polarities of each literal). These two updates are consecutively repeated for $T$ iterations.

$\mathbf{C_u}, \mathbf{L_u}$ denote two LayerNorm LSTMs (initialized randomly) with hidden states $C_h^{(t)} \in \mathbb{R}^{m \times d}, L_h^{(t)} \in \mathbb{R}^{2n \times d}$ respectively, and $\mathbf{L_{msg}}, \mathbf{C_{msg}}$ are multilayer perceptrons (MLPs) processing messages from literals and clauses. The last trained component is $\mathbf{L_{vote}}$, a voting MLP whose output is a single scalar for each literal. Edges of bipartite graph representation of the SAT formula are encoded in the bipartite adjacency matrix $M$ ($M(i,j)$ is 1 iff literal $l_i$ is in clause $c_j$). The flip operator swaps each pair of rows in matrix $L$, containing two polarities of the same literal.

To update a representation of each clause, the representations of literals contained in this clause are processed by the MLP $\mathbf{L_{msg}}$ and the resulting vectors are summed together and taken as input by the LSTM $C_u$.

We emphasize that for updating a representation of each literal, the process is analogous to the clause update, except that the LSTM takes as an input a concatenation of the summed messages from literals and the hidden-state representation of the literal of the same variable but opposite polarity (i.e., to update the hidden state of literal $x_i$, the LSTM takes as an input a concatenation of the aggregated message vector and a hidden state of literal $\bar{x}_i$ from the previous iteration).

At the end, the output of the model is a $2n$ dimensional vector, which is then averaged to a single logit on which a sigmoid activation cross-entropy is applied to compute the loss with respect to the ground truth label (SAT/UNSAT).

Our model is a simplified version of the described architecture, achieved by omitting two MLPs, namely $\mathbf{L_{msg}}, \mathbf{C_{msg}}$ and replacing $\mathbf{L_{vote}}$ with just a single linear layer. LayerNorm is removed from LSTM and the dimensionality of the hidden states is reduced to 16 from 128.

### A.3 DATASETS

#### A.3.1 RANDOM PROBLEMS

The generative model proposed by Selsam et al. (2019) samples formulas in sat/unsat pairs which differ only by a negation of a single literal in one clause. This is accomplished through the sequential sampling of clauses which are continuously added to the CNF formula until it becomes unsatisfiable. To create a new clause, the generative model first samples a small integer, $k$, and then randomly selects $k$ variables without replacement. Each selected variable is independently negated with a probability $0.5$ and the resulting literal is added to the clause. Satisfiability is determined by querying a solver right after the addition of a new clause. When the problem becomes unsatisfiable, it is paired with a satisfiable problem which is exactly the same except that in the last added clause, one literal is negated. The sampling of $k$ is designed to vary the size of clauses while avoiding an excessive number of two-literal clauses, which would simplify the problem on average.

#### A.3.2 STRUCTURED PROBLEMS

While many works evaluate NN-based SAT solvers on randomly generated problems, it is far more compelling to understand their performance on problems representing facets of human reasoning. The objective is to generate data that is reflecting various degrees of difficulty of Boolean reasoning. Since real-world problems often produce a large number of variables and clauses, which can be easily reduced by preprocessing, we uniformly reduce all the instances by unit propagation, which can be realized in polynomial time (see following paragraph).

**Unit propagation** is one of the simplest operation for propositional logic, which propagates unit clauses in a CNF $\phi$. The process consists of identifying a unit clause $\{l\} \in \phi$, then removing all clauses from $\phi$ that contain $\phi$ and removing the complementary literal $\bar{l}$ from all the other clauses. This process may create new unit clauses, which are then propagated in the same manner. If the process produces the empty clause (semantically equivalent to false), then the formula $\phi$ is unsatisfiable. We can say that a formula is *solved by* unit propagation if the unit operation derives the empty clause, or if all remaining clauses are unit clauses.

**Latin square** is an $n \times n$ grid of numbers $1..n$, where each number appears exactly once in each row and in each column. We generate SAT instances by partially filling the Latin square—the individual values in the partially filled Latin square are referred to as *hints*. Then, the task is to decide whether the given hints can be completed in to a full Latin square (similarly to the Sudoku puzzle). In order to generate interesting instances, we generate instances that have a unique solution and are minimal in the sense that removing any of the hints leads to multiple solutions. This is generated as follows. First generate a valid random Latin square and then start removing values of individual squares, at random, while a unique solution exists—this is checked by a SAT solver. The resulting formula consists of the rules of the Latin square and a set of unit clauses representing the hints. This process generates a satisfiable SAT instance with a unique solution. An unsatisfiable instance is generated by adding a single random hint incongruent with the unique solution.

**Sudoku** is a popular puzzle , which is in fact an extension of latin squares Where we add additional constraints on smaller squares (aka boxes). We consider the standard format where the puzzle is

composed of $3 \times 3$ boxes, which comprise $3 \times 3$ cells to be filled. We use the same method as in Latin squares to generate interesting puzzles.

**Logical circuits**   Are one of the main means of modeling in SAT. Indeed, they enable modeling digital systems but also represent a powerful intermediate language for modeling propositional problems. An important application of SAT are bit-vector problems of a fixed bit-width. To represent this type of reasoning, we generate problems of the form

$$c_1 * r_1 + c_2 * r_2 \neq c_3 \mod 2^n$$

We use the Model checker CBMC to convert these inequalities to CNF (Clarke et al., 2004).

## A.4   Visualizations of literal embeddings

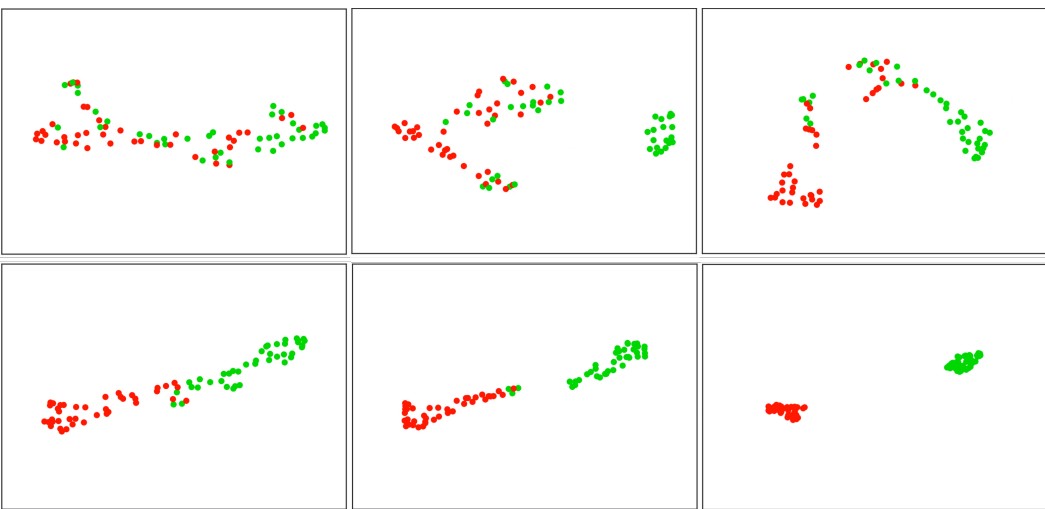

Figure 5: This figure shows evolution of the embeddigs of literals during the MP process. We selected 6 different time steps from the 30 time steps used for this example. The 16-dimensional vectors are projected to 2D by UMAP algorithm.

## A.5   Analysis of the Evolution of Literal Embeddings

To reinforce the assertion regarding the relationship between NeuroSAT and the SDP relaxation for MaxSAT, we tested whether the evolution of literal embeddings in NeuroSAT actually corresponds to an optimization process that tries to maximize the SDP objective 2.3.

We first sampled several hundred 2-CNF formulas and obtained the SDP objective function for each of them using the expressions mentioned in Appendix A.1. The objective function is a linear function of the Gram matrix $Y$ corresponding to the inner products between the unit vectors (representing the lifted variables and one vector $\mathbf{y_0}$ representing the value TRUE).

An SDP solver optimizes the matrix $Y$ while adhering to specified constraints (which ensure that the matrix can be obtained as a Gram matrix for some set of unit vectors). To observe the behavior of the same objective function with literal embeddings from NeuroSAT, we need to compute this Gram matrix $Y$ after each MP iteration of the GNN. If the evolution of these embeddings would correspond to an optimization process maximizing the objective, then we should observe an increase in this objective after each MP step.

When computing the matrix $Y$ from the NeuroSAT embeddings, two details must be taken into account. First, only the positive literals are taken into account as the objective function automatically assumes that the embeddings of negative literals are obtained by negation of the positive ones. Second, NeuroSAT does not explicitly represent the embedding $\mathbf{y_0}$ representing the value TRUE.

Therefore, we estimate it by averaging all literals that are assigned to TRUE in the extracted solution. Before computing the matrix $Y$, we also center all vectors to 0 and normalize them to unit vectors.

For each iteration $t$ of the GNN, we obtain the matrix $Y(t) = L^{(t)}L^{(t)T}$ where $L^{(t)}$ represents the matrix of centered and normalized positive literal embeddings with the estimated vector $\mathbf{y_0}$ (also normalized and centered) concatenated as its first row. In Figure 6 we show how the objective function changes after each iteration $t$ for 10 randomly selected instances. We also include the objective value obtained with a SDP solver as a reference.

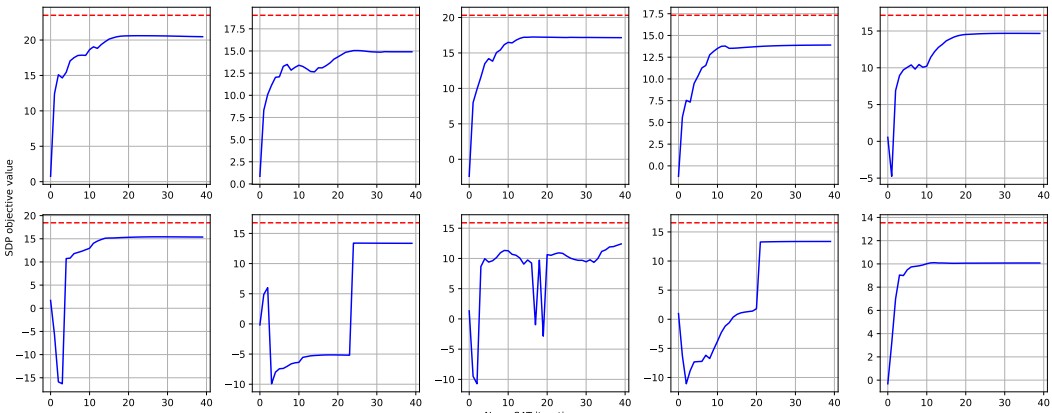

Figure 6: A plot showing how the SDP objective value computed from the NeuroSAT embeddings (in blue) changes after each iteration of MP. The horizontal red line represents the value of the same objective obtained with an SDP solver.

Figure 6 shows that the evolution of literal embeddings corresponds to an increase of the SDP objective value. It is also visible that there is a gap between the highest value achieved and the objective value obtained with an SDP solver. In Figure 7 (a), we plot a histogram of these gaps computed for all generated problems.

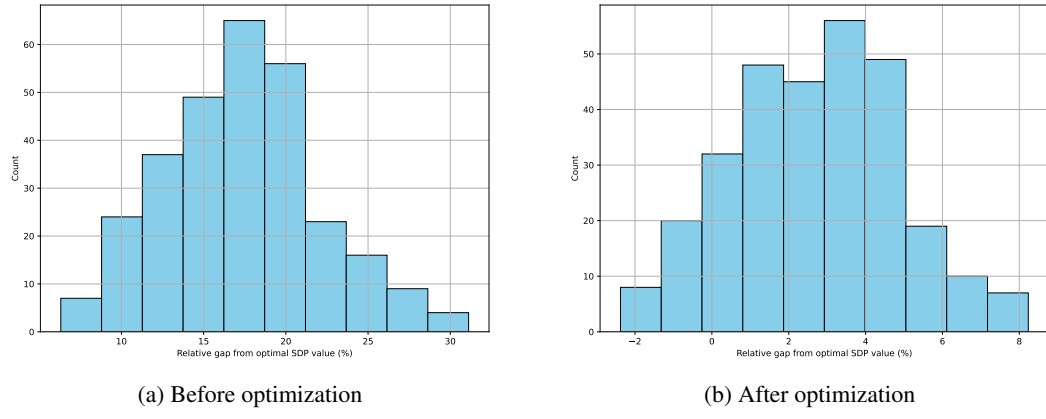

(a) Before optimization

(b) After optimization

Figure 7: Histograms of relative gaps between the final SDP objective value obtained from NeuroSAT embeddings and the same objective value obtained with an SDP solver. Plot (a) depicts results for the matrix $Y^{(t)}$ with $t = 40$. Plot (b) depicts results for the case where this matrix is further optimized by an SDP solver. A negative gap means that the subsequent optimization found a solution with better objective value the the SDP solver which was initialized randomly.

We hypothesized that the gap may be partially caused by the inappropriate choice of the vector $\mathbf{y_0}$. Therefore, we took the matrix $Y^{(t)}$ from the last step of MP, $t = 40$, and further optimized it using

a gradient-based SDP solver (implemented in PyTorch). This closed the gaps mentioned above in most instances, as visible in Figure 7 (b).

In Figure 8, we show how the entries in the matrix $Y^{(t)}$ change after further optimization for a random formula. As can be seen, the largest change in values occurs in the first row and in the first column, which correspond to inner products of each literal embedding with the vector $\mathbf{y_0}$. This supports our hypothesis that if we would be able to pick the vector $\mathbf{y_0}$ in a more optimal way, the gaps in Figure 6 would be smaller.

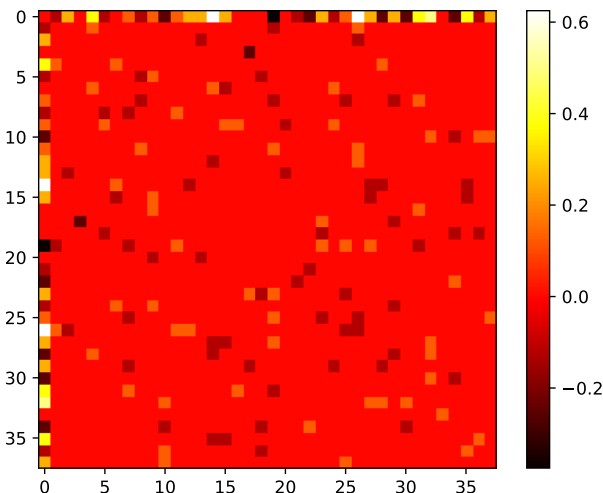

Figure 8: A heat map (for a single MAX-2-SAT instance) showing the change of values in the symmetric matrix $Y^{(t)}$ after further optimization. The largest change happens in the first row and in the first column, which correspond to inner products of each literal embedding with the vector $\mathbf{y_0}$ (corresponding to the value TRUE).

### A.6 SUPPLEMENTARY RESULTS FOR MORE DECIMATON STEPS

In Table 2, we show the effect of running the decimation process multiple times. On our test sets, the decimation process did not result in any further improvement when repeated more than twice. The hyperparameters are the same as for the experiments in Table 1 (with 16 random init. samples per formula in the first pass and 1 random init. sample for each subsequent pass). We note that we did not try to optimize the decimation threshold, which could lead to further improvements.

| Problem Type | #SAT problems | First pass | Second pass | Third pass |
|---|---|---|---|---|
| **SR(40)** | 5000 | 4442 (88.8 %) | 274 (5.4 %) | 35 (0.7 %) |
| **Latin Squares 9x9** | 200 | 186 (93 %) | 14 (7 %) | 4 (2 %) |
| **Latin Squares 8x8** | 200 | 196 (98 %) | 1 (0.5%) | 0 (0 %) |
| **Logical Circuits** | 344 | 319 (92.7 %) | 0 (0 %) | 0 (0 %) |
| **Sudoku 9x9** | 200 | 83 (46 %) | 11 (5.5 %) | 3 (1.5 %) |

Table 2: This table shows a number of problems solved after subsequent application of the decimation procedure.

