# OpenReview forum: "Revisiting GNNs for Boolean Satisfiability"
_ICLR.cc/2024/Conference — Submitted to ICLR 2024_

### Official Review · Reviewer_tuNF · 2023-10-28

**Soundness:** 2 fair
**Presentation:** 2 fair
**Contribution:** 2 fair
**Rating:** 5
**Confidence:** 4

**Summary:**

The paper primarily builds upon NeuroSAT and proposes several improvements to the original model, including the use of curriculum learning to speed up model training multiple initial assignments to embeddings, and decimation to enhance model accuracy.

**Strengths:**

Strengths:
1. The authors significantly accelerate training time by employing curriculum learning.
2. The model exhibits substantial improvements in accuracy compared to NeuroSAT.
3. The decimation measure, inspired by Belief Propagation (BP), is quite convincing.

**Weaknesses:**

Weaknesses:
1. Neither the samples nor the decimation techniques are subjected to ablation experiments.
2. The last sentence of the first paragraph in the introduction is not particularly convincing.
3. The sampling technique seems to enhance model accuracy solely by initializing values across multiple embeddings, and its relationship with SDP appears weak.

**Questions:**

Could you please clarify the nature of the initial embeddings? Are they generated randomly?

---

> ### Author Response · Authors · 2023-11-21
>
> Thank you for your feedback. We answer your comments and questions below.
>
> Comment 1:
> Neither the samples nor the decimation techniques are subjected to ablation experiments.
>
> Answer:
> Thank you for pointing this out, we extended the table with experiments (Table 1) so that the impact of decimation and sampling can be separated. Concretely, there is now a row for runs with 32 random initialization samples and no decimation. These runs will have the same number of samples as the runs in the first row with 16 samples in the first pass and 1 sample for each decimated problem in the second pass. One can, therefore, see the effect of decimation which is simply the difference between the number of solved problems in total between these two rows. The effect of sampling can be seen by comparing to the second row for runs with just 1 sample and no decimation.
>
> -----------------------------------------------
>
> Comment 2:
> `The first paragraph in the introduction is not particularly convincing.
>
> Answer:
> You probably meant this sentence: “The presented ideas should be easily generalizable to other combinatorial domains where these approximation algorithms could be applied.” We believe this should be the case, because SDP relaxation is a very general technique, and relaxations for many other combinatorial problems can be found in the literature. If the GNN is able to discover this relaxation, it should be, together with the ideas we presented, applicable to these other combinatorial domains. Nevertheless, we removed this sentence, as it is not essential and we did not have a space to expand on it.
>
> -----------------------------------------------
>
> Comment 3 and question 1:
> The sampling technique seems to enhance model accuracy solely by initializing values across multiple embeddings, and its relationship with SDP appears weak. Could you please clarify the nature of the initial embeddings? Are they generated randomly?
>
> Answer:
> The initial embeddings are random unit vectors. In Appendix A.5 we included results from other experiments which show that the iterations of the GNN can be viewed as optimization steps of the embeddings, optimizing the SDP objective value. Therefore, the intuition is that if these vectors are initialized in different positions, they can converge to a different local optimum. This is how the SDP solver behaves. Some initializations may be more lucky.

---

> ### Comment · Reviewer_tuNF · 2023-11-22
>
> Thank you for your response.
>
> Regarding the ablation study you have presented: including a scenario where only decimation is adopted could provide additional insights. Nevertheless, it is fine to see the results.

---

### Official Review · Reviewer_wK6B · 2023-10-29

**Soundness:** 2 fair
**Presentation:** 3 good
**Contribution:** 2 fair
**Rating:** 3
**Confidence:** 4

**Summary:**

This paper studies the Graph neural networks for combinatorial problems. In this work, the authors applied a curriculum training procedure, a decimation procedure and initial-value sampling. The authors claim that their proposed curriculum and optimization methods reduce training time by more than an order of magnitude and significantly increase the percentage of solved problems.

**Strengths:**

1. The paper is generally well-written, with clear explanations. It clearly introduces about the motivation of the optimizations and the methods applied.

2. Publicly available source code is provided to reproduce the results.

**Weaknesses:**

I have a number of comments regarding the experimental setup.

1. According to Section 5, the training instances are of very small scale. For the generated, random SAT instances, the number of variables are up to 40 variables. In fact, existing local search SAT algorithms are able to solve random satisfiable instances around phase-transition threshold with thousands of variables very efficiently. Hence, solving random instances with up to 40 variables are quite trivial.

2. After reading Appendix A.3.2, it seems that the authors also do not introduce the number of variables for those generated, structured instances (i.e., those instances generated from the domains of Latin squares, Sudoku, and logical circuits). Actually, it is widely recognized that modern CDCL SAT solvers can solve structured SAT instances with tens of thousands of variables. Could you please claim the numbers of variables for those generated, structured instances?

3. It seems that your proposed method can only handle satisfiable instances. Could you discuss the behavior of your proposed method when dealing with unsatisfiable instances?

4. The authors only compare their proposed method with NeuroSAT. However, CDCL solvers stand for the current state of the art in SAT solving. As a submission to a top-tier conference, lack a comparison against the real state of the art is unacceptable.

**Questions:**

Please see my comments that are listed in the Weaknesses.

**Details Of Ethics Concerns:**

I do not find ethics concern.

---

> ### Author Response · Authors · 2023-11-21
>
> Thank you for your comments. We believe that there was a misunderstanding of the purpose of our paper. We were not proposing a method that would be able to compete with mainstream SAT solvers. Our aim was to shed light on the process by which NeuroSAT learns to solve SAT instances. We clearly state that in the introduction: “Although these learned solutions lag far behind the solvers used in practice
> in terms of the size of problems they are able to solve, it is desirable to understand the process by which they arrive at their outputs.”
>
> We demonstrated the similarities of the trained NeuroSAT and two well-known approximation algorithms. Using these connections we proposed several improvements, one of which enables us to train NeuroSAT in order of magnitude shorter time, which can enable quicker experimentation with NeuroSAT.
>
> Regarding the comment: “As a submission to a top-tier conference, lack of a comparison against the real state of the art is unacceptable.” Please note that Selsam et al. published their paper about NeuroSAT at this conference and it was already cited more than 300 times. We believe that it is paramount to understand what kind of algorithms are GNNs able to learn, irrespective of how practically scalable these learned algorithms are.
>
> Many papers presented at this conference have similar flavors and show the behavior of the trained GNN only on toy examples (e.g. https://openreview.net/forum?id=rJxbJeHFPS , https://openreview.net/forum?id=hhvkdRdWt1F ).  These papers show how GNN are able to learn simple algorithms on small instances. It is clear that GNNs trained end-to-end are currently not able to compete with manually designed solvers. This kind of work fits well with ICLR as the conference is focused mainly on learning representations.
>
> Regarding unsatisfiable instances, we hoped it was clear from the text that our method cannot produce any guarantee of unsatisfiability. This is also the case for local search but it is still used in practice for instance within Cadical or Kissat. We also mentioned in the text that the solvers obtained from these approximation algorithms are incomplete. To make it clear, we now state it explicitly at the end of section 5.
>
> Regarding the number of variables of the structured formulas, they were included in Table 1, in the third column.

---

### Official Review · Reviewer_rKtv · 2023-10-31

**Soundness:** 3 good
**Presentation:** 3 good
**Contribution:** 3 good
**Rating:** 6
**Confidence:** 3

**Summary:**

This paper proposes two improvements over NeuroSAT, a very popular
message-passing NN for Boolean satisfiability (SAT).  These
improvements are inspired by two other approaches to (Max-)SAT:
Semidefinite Programming (SDP) relaxations and Belief Propagation
(BP). The first improvement is a form of curriculum learning, in which
the size of formulas and number of message-passing iterations is
increased throughout the training. This first improvement results in a
significantly faster training convergence. The second improvement is
twofold: 1) running in parallel NeuroSAT with multiple initializations
of the embedding vectors. 2) Once the model is trained, it is possible
to recover the notions of true/false values in the latent space. This
enables a decimation procedure during message passing, that is, early
fixing of the truth values if these get too close to true/false.
These two modifications result in more robust predictive performance.

**Strengths:**

- The presentation is good overall
- Well-motivated improvements over the existing work

**Weaknesses:**

- I have a few minor points on the presentation
- The experimental section does not address some important questions

---
Detailed comments:

  "Moreover, neural networks can potentially find solutions, which
  could lead to unexpected insight (Pickering et al., 2023; Davies et
  al., 2021)."

I don't understand this sentence. Is "solutions" in this context
referring to approximate algorithms for a target class of problems,
i.e. a trained model?  What unexpected insights are you hinting at?

  "Recently, Kyrillidis et al. (2020) demonstrates scenarios where
  solving a continuous relaxation formulation may provide benefits
  over solving the formula using standard solvers."

I would summarize these benefits.

  "[..] which has demonstrated the ability to exhibit nontrivial
  behaviors resembling a search in a continuous space, rather than
  mere classification based on superficial statistics."

What nontrivial behaviour are the authors referring to?

In Section 4: "Selsam et al. (2019) observes that for formulas that
the model correctly classified as satisfiable, the embeddings of
literals form two well-separated clusters."

This is already mentioned earlier in the text.

  "In Figure 4 in the Appendix, we recapitulate their visualization of
  embeddings with UMAP instead of PCA."

I was not able to connect the figure with the following text. Either
Fig. 4 is instrumental in understanding the following paragraphs and
should be moved to the main text, or it isn't and this sentence should
be removed. It is also unclear to me why you used UMAP instead of PCA.

It is not clear to me whether sampling multiple initializations and
decimation are two orthogonal improvements. If so, I would expect a
separate empirical evaluation for the two.

Given the nice performance improvements, I am left wondering if the
(augmented) NeuroSAT architecture is competitive in some settings with
other end-to-end approaches. This is not addressed in the experimental
section. Can we leverage curriculum learning to push the predictive
accuracy over 85% using a more expressive model? Can it also
better generalize to larger problems wrt the original NeuroSAT?

Reporting the inference time of the standard vs. decimation approaches
is necessary to the evaluation. I also wonder why only 2 passes of
decimation were evaluated. What happens if we do more?

**Questions:**

1) If multiple initializations and decimation are orthogonal improvements, can you provide ablation results for the two?
2) Can you provide a more throughout evaluation of decimation (i.e. with more than 2 passes)?
3) What is the runtime cost of your approach wrt standard NeuroSAT?
4) Does the augmented NeuroSAT method better generalize to larger instance?

---

> ### Author Response · Authors · 2023-11-21
>
> Thank you for your constructive feedback. Bellow are individual answers for your questions and some of your comments. For the comments we don’t reply to, changes were made in the text to make these points clearer):
>
> Comment:
> "In Figure 4 in the Appendix, we recapitulate their visualization of embeddings with UMAP instead of PCA."
> I was not able to connect the figure with the following text. Either Fig. 4 is instrumental in understanding the following paragraphs and should be moved to the main text, or it isn't and this sentence should be removed. It is also unclear to me why you used UMAP instead of PCA.
>
> Answer:
> Thank you for pointing this out. We included a simplified version of this figure into the main text. The figure shows how the embeddings evolve during the message-passing process. For satisfiable formulas, these embeddings create two well-separated clusters at the end. The text explains that it is possible to extract the solution of the formula by running a clustering algorithm on the embeddings produced by the last step of the MP process (for SAT formulas there should be 2 distinct clusters). One cluster will correspond to literals which will take the value TRUE and the other to literals which will take the value FALSE. We used UMAP as the clusters were more distinct in comparison to the visualization obtained by PCA.
>
> —-----------------------------------------------
>
> Comment and question 1:
> It is not clear to me whether sampling multiple initializations and decimation are two orthogonal improvements. If multiple initializations and decimation are orthogonal improvements, can you provide ablation results for the two?
>
> Answer:
> Thank you for pointing this out, we extended the table with experiments (Table 1) so that the impact of decimation and sampling can be separated. Concretely, there is now a row for runs with 32 random initialization samples and no decimation. These runs will have the same number of samples as the runs in the first row with 16 samples in the first pass and 1 sample for each decimated problem in the second pass. One can, therefore, see the effect of decimation which is simply the difference between the number of solved problems in total between these two rows. The effect of sampling only can be seen by comparing to the second row for runs with just 1 sample and no decimation.
>
> —-----------------------------------------------
>
> Comment:
> Given the nice performance improvements, I am left wondering if the (augmented) NeuroSAT architecture is competitive in some settings with other end-to-end approaches. This is not addressed in the experimental section.
>
> Answer:
> Our main goal in this paper was to shed light on the inner workings of trained NeuroSAT. Even though we show practical improvements, their main purpose was to support our claim that the trained GNN behaves as an SDP solver (which can be initialized with different values). Decimation on the other hand support the connection to the BP. The practical benefit of the curriculum is that one can iterate experiments with NeuroSAT faster.
> We are also interested in understanding how other end-to-end approaches are able to predict satisfiability (i.e. whether they are also doing something similar to an SDP solver) but we leave this direction for future work.
>
> —-----------------------------------------------
>
> Comment and question 4:
> Can we leverage curriculum learning to push the predictive accuracy over 85% using a more expressive model? Can it also better generalize to larger problems wrt the original NeuroSAT?
>
> Answer:
> The curriculum learning by itself does not seem to push the predictive accuracy. We saw that NeuraSAT with the curriculum can sometimes reach an accuracy of 87% (on the original test set) but we guess that NeuroSAT would also be able to reach this accuracy if trained long enough and with proper hyperparameters. We saw no difference in generalization to larger problems between the original NeuroSAT and the one trained with the curriculum.
>
> —-----------------------------------------------
>
> Question 2 and question 3:
> Can you provide a more throughout evaluation of decimation (i.e. with more than 2 passes)? What is the runtime cost of your approach wrt standard NeuroSAT?
>
> Answer:
> In Appendix A.6, we included a table with one more decimation step. The fourth pass did not solve any other problem in our test datasets. We will study this more closely in future work.
> We did not optimize the runtime of the decimation procedure. We decimate and obtain a new prediction for each sample separately. If we have 16 samples for which we obtain predictions in parallel and for each of the 16 decimated formulas we produce 1 sample, then this should take approximately 16x more time than without the decimation. Decimation could probably be parallelized but as these end-to-end approaches are currently not used in practice we did not explore such optimizations.

---

> > ### Comment · Reviewer_rKtv · 2023-11-22
> > **Response to authors**
> >
> > Thank you for your response, I think I have a better grasp of the purpose and significance of your work now.
> > The changes to the manuscript are also welcome.

---

### Official Review · Reviewer_mpUf · 2023-11-01

**Soundness:** 3 good
**Presentation:** 4 excellent
**Contribution:** 3 good
**Rating:** 6
**Confidence:** 3

**Summary:**

This paper proposes enhancements for the training and inference procedure of Graph Neural Networks (GNNs) that are trained to predict solutions of combinatorial problems, with a focus on Boolean Satisfiability (SAT). The proposed optimizations include a curriculum training procedure, a novel loss function, and a dynamic batching strategy. The idea is inspired by the possible connection of the behavior of GNN and two algorithms: Belief Propagation and Semidefinite Programming Relaxations. These enhancements significantly reduce training time and increase the percentage of solved problems. The paper also provides a comprehensive review of related work in the context of GNNs and Boolean Satisfiability.

**Strengths:**

The problem studied in this paper is fundamental: what does a GNN learn? Does it devise a new algorithm? Combinatorial problems are perfect objections for conducting those studies as they are well-studied and we already know a bunch of algorithms. NeuroSAT is a well-known work in the application of GNN on combinatorial problems. What algorithm NeuroSAT really learns has not been fully understood. Therefore, the behavior of NeuroSAT is of great interest.

I like the algorithmic part of this work, which greatly improves the training efficiency. This paper also simplifies the structure of NeuroSAT, which may make it easier for future work to investigate its behavior.

The paper is well-written and easy to follow. The introduction and preliminary sections give comprehensive context and background.

Despite the over-claimed connection between NeuroSAT and SDP/MP, I still like the direction of this work. I am happy to change my evaluation if my concerns can be addressed.

**Weaknesses:**

The paper claims to reveal the similarity of GNN and Belief propagation. However, there is little convincing evidence of those similarities, in my opinion. It is mentioned in the paper that:

"For satisfiable formulas, this happens when the vectors form two well-separated clusters, which makes the whole
process qualitatively similar to the optimization of the SDP relaxation described in Section 2.3."

The vectors forming two well-separated clusters, while interesting, is not strong evidence that NeuroSAT is similar to SDP. There may be other algorithms for SAT based on lifting to high-dimension vectors that also obey this behavior. Either stronger evidence (e.g. NeuroSAT is optimizing some quadratic objective) should be revealed or the statement of NeuroSAT & SDP should be removed. It would be great to see more experiments for the behavior of NeuroSAT, besides showing the efficiency of the new network structure with the curriculum.

**Questions:**

Can more evidence be discovered for the similarity of NeuroSAT and SDP/MP?

---

> ### Author Response · Authors · 2023-11-21
>
> Thank you, we appreciate your constructive feedback. Based on your comments we ran several experiments to explore the connection to SDP further and included the results in the appendix A.5. Here we summarize our findings for 2-SAT formulas:
>
> First, we visualized what happens with the SDP objective if we compute it from the embeddings of literals obtained from NeuroSAT after each MP iteration. Our intuition was that each message-passing iteration would increase the SDP objective.
>
> The objective is a linear function of the Gram matrix (matrix with all pairs of inner products) of the literal embeddings together with the embedding for the additional variable representing the value TRUE. As this additional variable is not explicitly represented in NeuroSAT, we obtain it by averaging all embeddings of literals that will be assigned to value TRUE in the final solution. Before computing the Gram matrix and the SDP objective value, we also center all embeddings to 0 and normalize them to unit vectors. As can be seen in Figure 6 in the appendix, the SDP objective value is mostly increasing after each MP iteration.
>
> There is a visible gap between the solution from NeuroSAT and the solution found by the SDP solver. In the text, we explain that this gap is largely caused by suboptimally choosing the vector representing the value TRUE. Therefore, the evolution of the literal embeddings in NeuroSAT can be seen as an optimization process of the SDP objective (modulo the centering and normalization).
>
> We believe that it is valuable to point to these connections as the SDP relaxation for MaxSAT is well-known and can provide ideas for experiments. To the best of our knowledge, it is the only theoretically understood algorithm for MaxSAT that lifts the literals to high dimensional vectors.
>
> To not over-claim the connections, we also added the following two sentences at the end of the introduction.
> “We also emphasize that we do not derive any formal statements regarding these connections. We demonstrate that the behavior of the trained GNN is qualitatively similar to these approximation algorithms and that using insights from these approximation algorithms can lead to practical improvements.”

---

> > ### Comment · Reviewer_mpUf · 2023-11-22
> >
> > Thank you for the response and the results! With the changes and clarifications, I am willing to increase my score.

---

### Meta-Review · Area_Chair_z7xD · 2023-12-06

**Metareview:**

This work introduces a curriculum training procedure of GNN to predict solutions to SAT problems. A novel loss function and a dynamic batching strategy are also provided to improve the performance. The reviewers acknowledge its presentation and technical soundness. However, the main concerns lie in the experimental setups and lack of performance on unsatisfiable instances and comparison with related works. Taking the mixed opinion into account, I opt for rejection.

**Justification For Why Not Higher Score:**

Please see the meta-review.

**Justification For Why Not Lower Score:**

N/A

---

### Decision · Program_Chairs · 2024-01-16

Reject